# Driving Intention Inference Based on a Deep Neural Network with Dropout Regularization from Adhesion Coefficients in Active Collision Avoidance Control Systems

Yufeng Lian [1], Jianan Huang [1], Shuaishi Liu [1], Zhongbo Sun [1], Binglin Li [1] and Zhigen Nie [2,*]

[1] School of Electrical and Electronic Engineering, Changchun University of Technology, Changchun 130012, China; lianyufeng_1982@126.com (Y.L.); 2202004028@stu.ccut.edu.cn (J.H.); liushuaishi@ccut.edu.cn (S.L.); sunzhongbo@ccut.edu.cn (Z.S.); libinglin@ccut.edu.cn (B.L.)

[2] Faculty of Transportation Engineering, Kunming University of Science and Technology, Kunming 650500, China

* Correspondence: niezhigen@126.com; Tel.: +86-871-65920069

**Abstract:** Driving intention, which can assist drivers to avoid dangerous emergence for the advanced driver assistant systems (ADAS), can be hardly described accurately for complex traffic environments. At present, driving intention can be mainly obtained by deep neural networks with neuromuscular dynamics and electromyography (EMG) signals of drivers. This method needs numerous drivers' signals and neural networks with a complex structure. This paper proposes a driving intention direct inference method, namely direct inference from the road surface condition. A driving intention safety distance model based on a deep neural network with dropout regularization was built in an active collision avoidance control system of electric vehicles. Driving intention can be inferred by a deep neural network with dropout regularization from adhesion coefficients between the tire and road. Simulations using rapid control prototyping (RCP) and a hardware-in-the-loop (HIL) simulator were performed to demonstrate the effectiveness of the proposed driving intention safety distance model based on a deep neural network with dropout regularization. The proposed driving intention safety distance model can guarantee the safe driving of electric vehicles.

**Keywords:** active collision avoidance system; driving intention inference; deep neural network; dropout regularization; electric vehicles

## 1. Introduction

### 1.1. Motivation

Vehicle safety is always the primary task and the premise of vehicle research and design in the intelligent transportation system (ITS) [1]. Numerous studies focus on active collision avoidance control systems to improve vehicle safety for the advanced driver assistant system (ADAS). It mainly contains the following modules: perception and processing of driving information [2], safety distance calculation and safety state judgement [3], vehicle dynamics and control [4], and control command execution and power transmission [5]. The overall system architecture of the active collision avoidance control system can be summarized as shown in Figure 1. The safety distance model plays an important role in vehicle safety and traffic utilization for the active collision avoidance control system. Safety distance models can be divided into three types [2], namely the kinematics-based safety distance model, vehicle-to-vehicle distance-based safety distance model, and driver's characteristics-based safety distance model. Although the safety distance models have been improved continuously, there are still some problems to be solved, such as the low accuracy, the lock of drivers' characteristics, and the poor adaptability of a complex traffic environment.

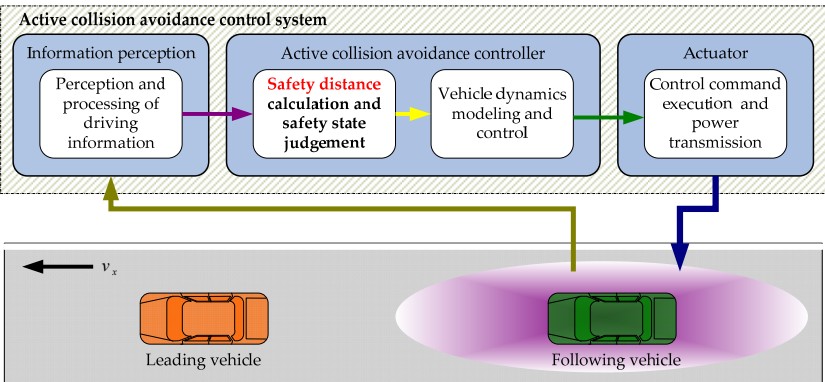

**Figure 1.** The overall system architecture of active collision avoidance control system.

### 1.2. Related Works

In recent years, the driver's characteristics-based safety distance model was given more and more attention by researchers for active collision avoidance control systems. For the driver's characteristics-based safety distance model, existing studies mainly focus on driving intention inference [6,7]. Based on different criteria, driving intention inference can be divided two aspects: maneuver input inference and tactical behavior decision inference [6]. Maneuver inputs can be inferred for low-level controllers and tactical behavior decisions can be inferred for high-level controllers in the hierarchical active collision avoidance control system.

This paper mainly focuses on the driving intention inference of tactical behavior decisions. Most approaches were investigated and the technology of the complex sensor fusion and data coordination was used for obtaining driving intention [6]. Driving intention inference has been grouped into three parts [8]: The first part is physics-based intention inference, which describes vehicles with dynamic and kinematic models by the laws of physics. Dynamic models, which can describe vehicle motion with Lagrange mechanics, can become extremely large and involve many internal parameters of the vehicle. By contrast, kinematic models are simpler and use more extensive data for trajectory prediction. Physics-based intention inference is only suitable for short-term motion inference, therefore this kind of model is unable to anticipate any change in the motion of the car without drivers' characteristics. The second part is maneuver-based intention inference, which describes vehicles as independent maneuvering entities. It can obtain more relevant and reliable trajectories than the ones obtained from physics-based intention inference. The Hidden Markov Model (HMMs) and the model based on fuzzy reasoning are two widely used methods. An intention predication model based on attribute-driven Hidden Markov Model Trees is proposed for intention prediction [9]. An efficient recognition approach based on Non-linear Polynomial Regression (NPR) and the Recurrent Hidden Semi-Markov Model (R-HSMM) is proposed to accurately recognize the driver lane-change intention in the early stage [10]. HMMs can only focus on current driving behaviors rather than past driving behaviors, although they are widely used for driving intention inference. Fuzzy reasoning methods are a class of experience models, therefore they can obtain a well-quantified driving intention. A fuzzy reasoning-based longitudinal minimum safety distance model was designed with the information on drivers' intentions and driving circumstance [11]. A fuzzy logic inference system has been applied to identify driving intention for hybrid vehicles. The membership functions and rules of the fuzzy logic inference system are built for intention identification and the simulation is done in different driving conditions [12]. Driving intention inference is affected by the compatibility of different membership functions and fuzzy rules comparing to HMMs. In addition to that, Support Vector Machines (SVMs) [13], Artificial Neural Network (ANN) models [14], and the Bayesian Network (BN) [15] have been developed to infer driving intention. Maneuver-based intention inference disregards the dependence between vehicles, resulting in the

wrong judgement of situations. The third part is interaction-aware intention inference, which describes vehicle motion combined with the motion of other vehicles. Compared to maneuver-based intention inference, it can infer driving intention more reasonably resulting from the dependence between vehicles. Dynamic Bayesian Networks (DBN) are exploited widely for interaction-aware intention inference [16–18]. Coupled Hidden Markov models (CHMMs) can also be used with pairwise dependencies between other vehicles [19,20]. Apart from that, the development of deep neural networks provides a good solution for driving intention inference. The task of driving behavior recognition can be regarded as a multi-class classification problem. A multi-stream Convolutional Neural Network (CNN) was employed to extract multi-scale features by filtering images with receptive fields of different kernel sizes and the final decision is generated with different fusion strategies for driving behavior recognition [21]. A novel data-driven modelling methodology was proposed for the lateral stability description of articulated steering vehicles and a Recurrent Neural Network (RNN) model was built to accurately quantify vehicle lateral stability [22]. A long short-term memory (LSTM) network, which is a time cycle neural network, was used to evaluate real-time bus riding comfort and provide driving suggestions [23]. A CNN-LSTM method was proposed to meet the prediction requirements and provide an effective method for the safe operation of unmanned systems [24]. The information on other vehicles is taken into account in interaction-aware intention inference and the obtained driving intention is more consistent with the actual driving conditions. It also brings some disadvantages considering that multi-information fusion results in increased complexity of the model structure, increased computational load, and poor real-time performance.

This paper proposes a driving intention safety distance model based on a deep neural network with dropout regularization. The deep neural network is designed to obtain driving intention directly from the adhesion coefficients between the tire and road. Dropout regularization is utilized to train deep neural networks to prevent the network overfitting problem. The remaining paper is organized as follows: Section 2 proposes a driving intention safety distance model based on a deep neural network with dropout regularization. The deep neural network is validated with random data. Section 3 designs an active collision avoidance control system based on the driving intention safety distance model. Simulation results are presented and discussed. Finally, there is a conclusion regarding research and future works in Section 4.

## 2. Deep Neural Network-Based Driving Intention Safety Distance

### 2.1. Driving Intention Safety Distance Modeling

This paper extends and improves the authors' previous work. The safety distance model can be expressed as follows [2,5]:

$$d_0 = \frac{k_d c}{d + \frac{\varphi_{fl} + \varphi_{fr} + \varphi_{rl} + \varphi_{rr}}{4}} \qquad (1)$$

where $\varphi_{fl}, \varphi_{fr}, \varphi_{rl}$, and $\varphi_{rr}$ are the adhesion coefficients of the four tires, respectively. They can describe the road conditions of vehicle driving. In this work, the safety distance model is modeled with the average of the adhesion coefficients for four tires to evaluate and synthetically describe the whole road condition of the driving vehicle. Constants $c$ and $d$ are safety distance model parameters [2,5]. $k_d > 0$ is used to describe driving intention, which is associated with different road conditions, namely the average of the adhesion coefficients. In previous work, $k_d$ was set with a fixed constant in the whole driving process [2,5]. Although it can ensure vehicle safety, it is so conservative that road utilization is reduced due to the excessive safety distance. Indeed, $k_d$ should be a non-linear function, which is associated with the average of the adhesion coefficients. Therefore, it should be improved as a function that varies with the average of the adhesion coefficient to conform to the drivers' driving characteristics.

### 2.2. Driving Intention Deep Neural Network

In terms of deep learning, a hybrid-learning-based driving intention network was employed with neuromuscular dynamics and electromyography (EMG) signals of drivers collected from far-reaching studies [6]. The predication scheme needs to extract some EMG signals from the upper limb, such as the clavicular portion, deltoid anterior, deltoid posterior, triceps, and so forth. With the large amount of data processing and complex structure of the deep neural network, the inferred driving intention would deviate from the actual intention. In this work, a deep neural network is designed from road conditions rather than drivers.

As is known to all, the safety distance model mainly depends on road conditions. Therefore, the road condition is regarded as the only input of the deep neural network and the driving intention is regarded as the output of the deep neural network, which directly reflects the relationship between the road condition and driving intention. To simplify the deep neural network structure, the biases of the nodes are regarded as 0, and the weights are considered in this work. The deep neural network contains 1 input node, 3 hidden layers, and 1 output node, as shown in Figure 2. Each hidden layer has 20 nodes and the sigmoid function is used as the activation function in the hidden layers. The rectified linear unit (ReLU) function is used as the activation function in the output layer. Apart from that, dropout regularization, which can train only some of the randomly selected nodes rather than the entire neural network, is used to prevent the deep neural network from overfitting. It is the most representative solution to prevent overfitting [25]. Some nodes are randomly selected at a certain percentage and their outputs are set to be 0 to deactivate the nodes.

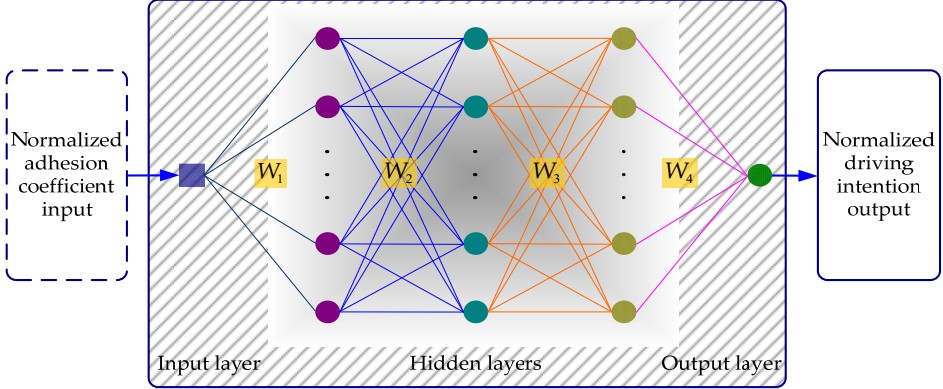

**Figure 2.** Deep neural network with 3 hidden layers.

### 2.3. Simulation Results and Analysis

Based on the experimental results of Yan's work [26] and (1) [2], driving intention $k_d$ can be calculated as training data, which is between 130 and 110, for training the deep neural network under different driving economies, namely the High Way Fuel Economy Test (HWFET) and Urban Dynamometer Driving Schedule (UDDS). The training data of driving intention must to be normalized to the data between 0 and 1 before training the deep neural network. Since the range of adhesion coefficients is between 0 and 1, the training data of driving intention can be directly regarded as the input of the deep neural network, which can be seen in part of the dotted line. In the deep neural network, 3 hidden layers are all designed with 20 nodes, namely $W_1$ is a matrix by 20 rows and 1 column, $W_2$ is a matrix by 20 rows and 20 columns, $W_3$ is a matrix by 20 rows and 20 columns, and $W_4$ is a matrix by 1 row and 20 columns. The epoch of the deep neural network is set as 4000. $W_1$, $W_2$, $W_3$, and $W_4$ are set to random numbers between $-1$ and 1. The learning time of the deep neural network is 73.0213 s. The output of the deep neural network should be also normalized to the actual driving intention range before providing driving intention information for an active collision avoidance control system.

The deep neural network is validated with 20,000 random data and the random data validation curve is shown in Figure 3. In total, 20,000 random data are all between 0 and 1, and the predictive values of the deep neural network are between 110 and 120. The reaction time of the deep neural network is 0.97037 s. Since biases are ignored in this neural network, the driving intention can be inferred quickly. This reaction time, which is faster than drivers' reaction time, can timely control vehicle safety driving with the active collision avoidance control system. It is suitable for describing drivers' characteristics. Predictive values deviate slightly from the training data for this deep neural network, resulting from ignored biases. Although there are deviations between the correct output and actual output, the designed deep neural network could infer driving intention. When the adhesion coefficient is small and the road condition is poor, the value of the driving intention is large enough to increase the vehicle-to-vehicle distance to guarantee vehicle safety. On the contrary, when the adhesion coefficient is large and the road condition is good, the value of the driving intention becomes smaller to reduce the vehicle-to-vehicle distance to improve the road utilization rate.

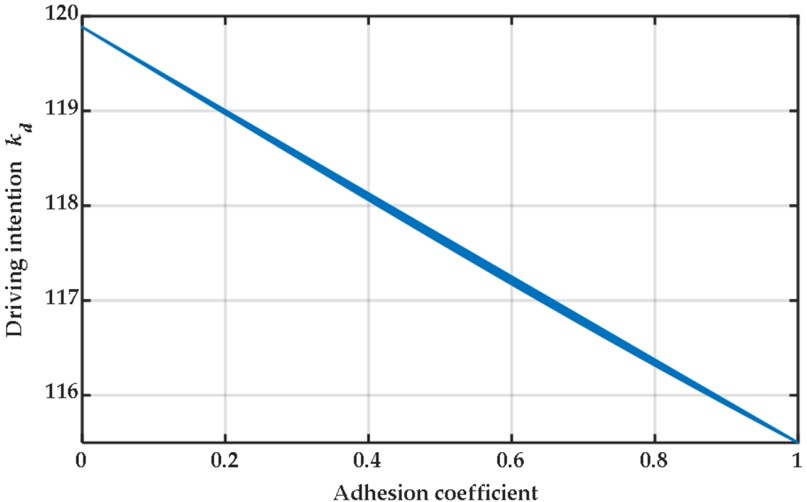

**Figure 3.** Random data validation for deep neural network.

## 3. Active Collision Avoidance Control System

### 3.1. Active Collision Avoidance Control System Structure

A four-in-wheel-motor-driven electric vehicle (FIWMD-EV) simulation platform is presented and shown in Figure 4. MicroAutoBoxII;, which is a rapid control prototyping (RCP) device, is used as the simulator of active collision avoidance controllers. The dSPACE HIL simulator is used as the simulator of vehicle dynamics.

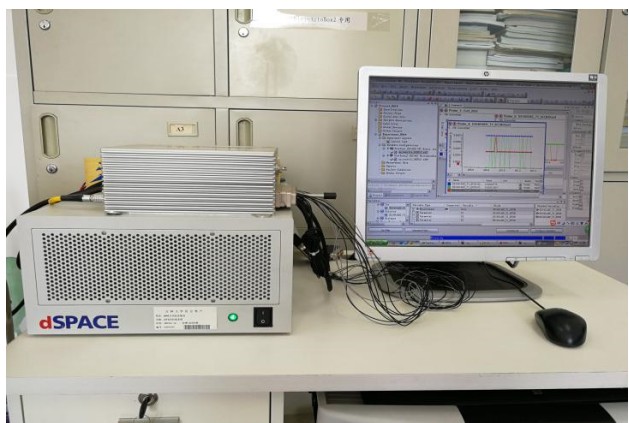

**Figure 4.** Hardware platform of the active collision avoidance control system.

The structure diagram of the active collision avoidance control system is shown in Figure 5 and the following vehicle velocity $v_1$ and vehicle acceleration $a_1$ are obtained from CarSim 8.02. Leading vehicle velocity $v_2$ is obtained from ADVISOR 2002. The braking-critical distance and warn-critical distance, which are used to participate in vehicle safety state judgement; upper and lower-level controllers; and dynamics were proposed and presented in authors' previous work [2,5,26], and are employed again in this paper. Apart from that, different driving roads are set by road characteristic parameters in LuGre dynamic friction models for the four tires. A LuGre dynamic friction model is expressed as follows [27]:

$$
\begin{cases}
\dot{z} = v_r - \theta \frac{\sigma_0 |v_r|}{g(v_r)} z - \kappa r |w_e| z \\
F_{xi,j} = \mu_{i,j} F_{zi,j} = (\sigma_0 z + \sigma_1 \dot{z} + \sigma_2 v_r) F_{zi,j} \\
g(v_r) = \mu_c + (\mu_s - \mu_c) e^{-\left|\frac{v_r}{v_s}\right|}
\end{cases}
, \quad \left( \begin{array}{l} i = 1 : front, 2 : rear \\ j = 1 : left, 2 : right \end{array} \right) \tag{2}
$$

where $v_r = w_e r - v_1$ is the relative velocity between the wheel velocity and following vehicle velocity; $r$ is the wheel radius; $\theta$ is the road characteristic parameter; $v_s$ is the Stribeck relation velocity; $F_x$ is the longitudinal tire force; $F_z$ is the tire normal force; $z$ is the internal friction state of the tire; $\sigma_0$ is the normalized rubber longitudinal-lumped stiffness; $\sigma_1$ is the normalized rubber longitudinal-lumped damping; $\sigma_2$ is the normalized viscous relative damping; $\mu_c$ is the normalized coulomb friction; and $\mu_s$ is the normalized static friction. According to (2), the maximum steady-state value of the friction coefficient, which can be represented by the adhesion coefficients of complex roads, can be obtained as follows:

$$
\mu_{ssm} = \frac{\sigma_0 v_{rm}}{\frac{\theta \sigma_0 |v_{rm}|}{g(v_{rm})} + \kappa |v_{rm} + v_1|} + \sigma_2 v_{rm} \tag{3}
$$

where $v_{rm}$ can be calculated, as mentioned in [5,27]. In this work, $\theta$ is set with changing values to represent complex roads.

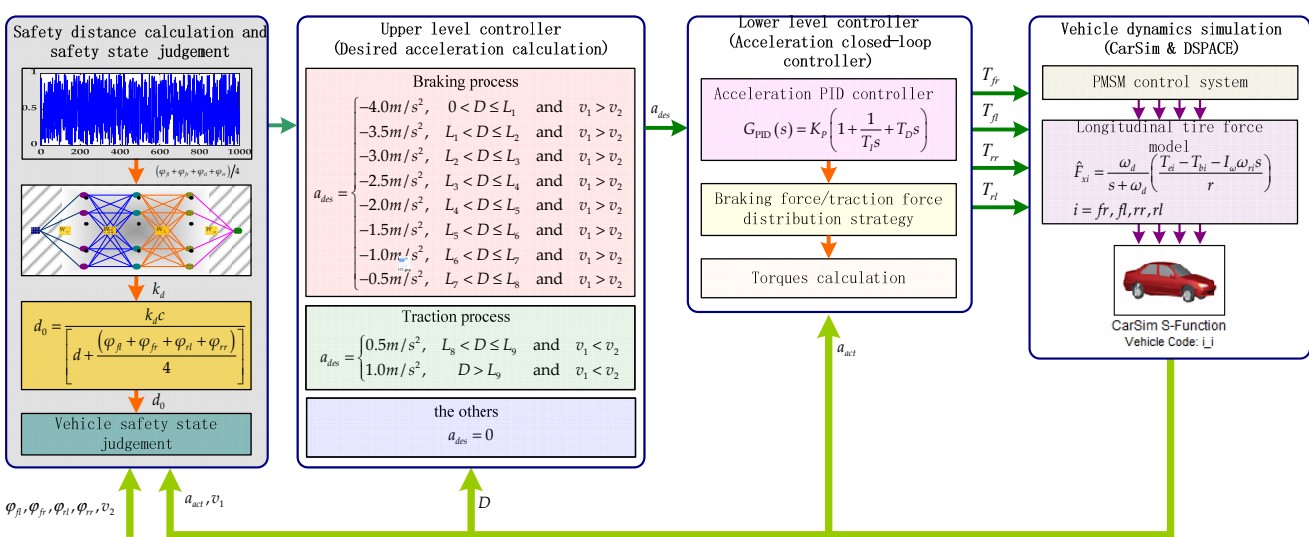

**Figure 5.** Structure diagram of the active collision avoidance system.

### 3.2. Simulation Results and Analysis

According to literature [2], constants $c$ and $d$, which are the parameters in (1), are determined with general values [2,5], namely $c = 2$ and $d = 0.3$. The safety distance model can be given by (4). A LuGre dynamic friction model and vehicle dynamics parameters are listed in Tables 1 and 2, respectively. Simulation experiments were performed under a snow and ice road, and a dry asphalt–snow–ice-mixed road, which was realized by setting road characteristic parameter $\theta$ in UDDS and HWFET. According to literature [27], road

characteristic parameter $\theta$ could be set to 1, which can represent a dry asphalt road, and road characteristic parameter $\theta$ could be set to 6, which can represent a snow and ice road. Therefore, the road conditions can be set by road characteristic parameter $\theta$ in the LuGre dynamic friction model. UDDS and HWFET were used to represent the velocity of the leading vehicle in simulation experiments.

$$d_0 = \frac{2k_d}{0.3 + \frac{\varphi_{fl} + \varphi_{fr} + \varphi_{rl} + \varphi_{rr}}{4}} \tag{4}$$

**Table 1.** LuGre dynamic friction model parameters.

| Parameter | Value | Parameter | Value |
|-----------|-------|-----------|-------|
| $\mu_c$ | 0.3 | $L_{tire}$ | 0.2 m |
| $\mu_s$ | 1.4 | $\sigma_0$ | 150 1/m |
| $v_s$ | 1.5 m/s | $\sigma_1$ | 4 s/m |
| $\kappa$ | $\frac{7}{6}L_{tire}$ | $\sigma_2$ | 0.01 s/m |

**Table 2.** Vehicle dynamics parameters.

| Parameter | Value | Parameter | Value |
|-----------|-------|-----------|-------|
| $A_f$ [1] | $1.6 + 0.00056(m - 765)$ | $h_{center}$ [8] | 0.5 m |
| $C_d$ [2] | 0.3 | $a$ [9] | 1.04 m |
| $D(0)$ [3] | 100 m | $b$ [10] | 1.56 m |
| $v_1(0)$ [4] | 0.0 m/s | $g$ | 9.8 m/s$^2$ |
| $v_{wind}$ [5] | 1 km/h | Engine type | Permanent magnet synchronous motor (PMSM) |
| $\rho$ [6] | 1.225 kg/m$^3$ | Maximum power | 10.7 kW |
| $r$ | 0.313 m | Maximum speed | 1500 rpm |
| $m$ [7] | 1159 kg | Maximum torque | 340 Nm |

[1] Frontal area of the vehicle. [2] Aerodynamic drag coefficient. [3] Initial vehicle-to-vehicle distance. [4] Initial velocity of following vehicle. [5] Wind velocity. [6] Mass density of air. [7] Vehicle mass. [8] Height of the center of gravity (CG). [9] Distance from CG to front axle. [10] Distance from CG to rear axle.

### 3.2.1. UDDS Simulation Experiments

UDDS was selected as the simulation urban road condition in this work. The cycle time was 1367 s; distance traveled was 11.99 km; maximum speed was 91.28 km/h; average speed was 31.51 km/h; maximum acceleration was 1.48 m/s$^2$; maximum deceleration was $-1.48$ m/s$^2$; and number of stops was 17 [2,5,28].

(1) Snow and ice road simulation experiments

Road characteristic parameter $\theta$ can be set to the same values, namely $\theta = 6$, in order for the left and right tires to represent the snow and ice road condition. As seen in Figure 6a, the road conditions of the left and right tires were the same. The average adhesion coefficient, which is shown in Figure 6b, ias calculated by four adhesion coefficients from the LuGre dynamic friction models of the four tires. With the average adhesion coefficient, driving intention $k_d$ can be inferred by the deep neural network. For a FIWMD-EV, the LuGre dynamic friction model can calculate the adhesion coefficient with the road characteristic parameter $\theta$ and PMSM speed. Hence, the calculated adhesion coefficient can vary between 0.0 and 0.2 due to the change of the PMSM speed. The average adhesion coefficient provides road surface information for the deep neural network to infer driving intention. Compared to the change trends in Figure 6b,c, the variation trend of driving intention is opposite to that of the average adhesion coefficient. That is to say, when the average adhesion coefficient is small, the driving intention is large, and vice versa; it conforms to drivers' driving habits. Acceleration closed-loop curves are shown in Figure 6d and actual acceleration can basically track the desired acceleration in the whole driving process. The actual acceleration curve fluctuated once at about 170 s, which was mainly caused by

the average adhesion coefficient fluctuation, resulting in the fluctuation of driving intention. The fluctuation can immediately disappear due to acceleration closed-loop control, namely the adjustment action of the lower-level controller. Longitudinal velocities of the leading and following vehicles are shown in Figure 6e, and the vehicle-to-vehicle distance and safety distance are shown in Figure 6f. Actual vehicle-to-vehicle distance is basically greater than the safe distance and safe distance is greater than 400 m. This demonstrates that electric vehicles can drive safely and the proposed driving intention safety distance model is reasonable, feasible, and effective in UDDS.

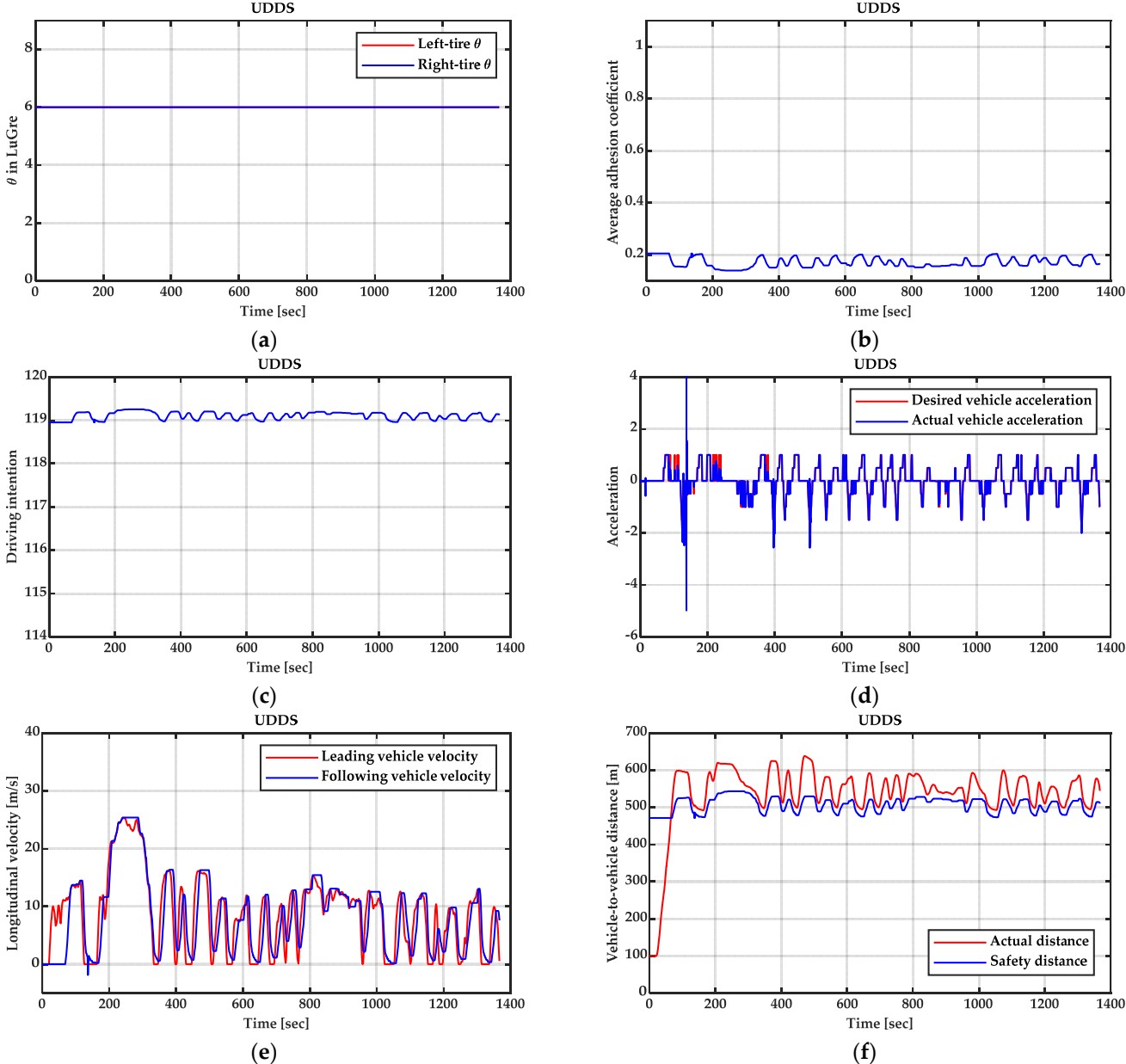

**Figure 6.** Snow and ice road experiments in UDDS with deep neural network. (**a**) Road characteristic parameter of left and right tires. (**b**) Average adhesion coefficient. (**c**) Driving intention. (**d**) Acceleration of following vehicle. (**e**) Longitudinal velocities of leading and following vehicles. (**f**) Vehicle-to-vehicle distance and actual distance.

(2) Dry asphalt–snow–ice-mixed road simulation experiments

Road characteristic parameter $\theta$ can be set to different values in order for the left and right tires to represent the different road conditions. As seen in Figure 7a, the road

conditions of the left and right tires had the same situation but also different situations. The road conditions of the left and right tires included a dry asphalt road as well as a snow and ice road. Therefore, they can represent all the road conditions encountered by tires. The average adhesion coefficient, which is shown in Figure 7b, was calculated by the four adhesion coefficients from the LuGre dynamic friction models of the four tires. With the average adhesion coefficient, driving intention $k_d$ can be inferred by the deep neural network. The calculated adhesion coefficient can vary between 0.0 and 1.0 due to the change of the PMSM speed. Figure 7c,d show the adhesion coefficients of the four tires. The adhesion coefficients of the left tires are the same as that of the right tires. Furthermore, the values of the adhesion coefficients of the four tires cover all road conditions, namely a dry asphalt road as well as a snow and ice road. They can provide adequate road surface information for the deep neural network to infer driving intention. Compared to the change trends in Figure 7b,e, the variation trend of the driving intention is opposite to that of the average adhesion coefficient. That is to say, when the average adhesion coefficient is small, the driving intention is large, and vice versa; it also conforms to drivers' driving habits. Acceleration closed-loop curves are shown in Figure 7f and actual acceleration can basically track the desired acceleration in the whole driving process. The actual acceleration curve fluctuated once at about 180 s, which was mainly caused by the road condition changes of the left and right tires. The fluctuation can immediately disappear due to acceleration closed-loop control. Longitudinal velocities of the leading and following vehicles are shown in Figure 7g, and the vehicle-to-vehicle distance and safety distance are shown in Figure 7h. The actual vehicle-to-vehicle distance is greater than the safe distance and the safe distance is greater than 180 m. This also demonstrates that electric vehicles can drive safely and the proposed driving intention safety distance model is reasonable, feasible, and effective in UDDS.

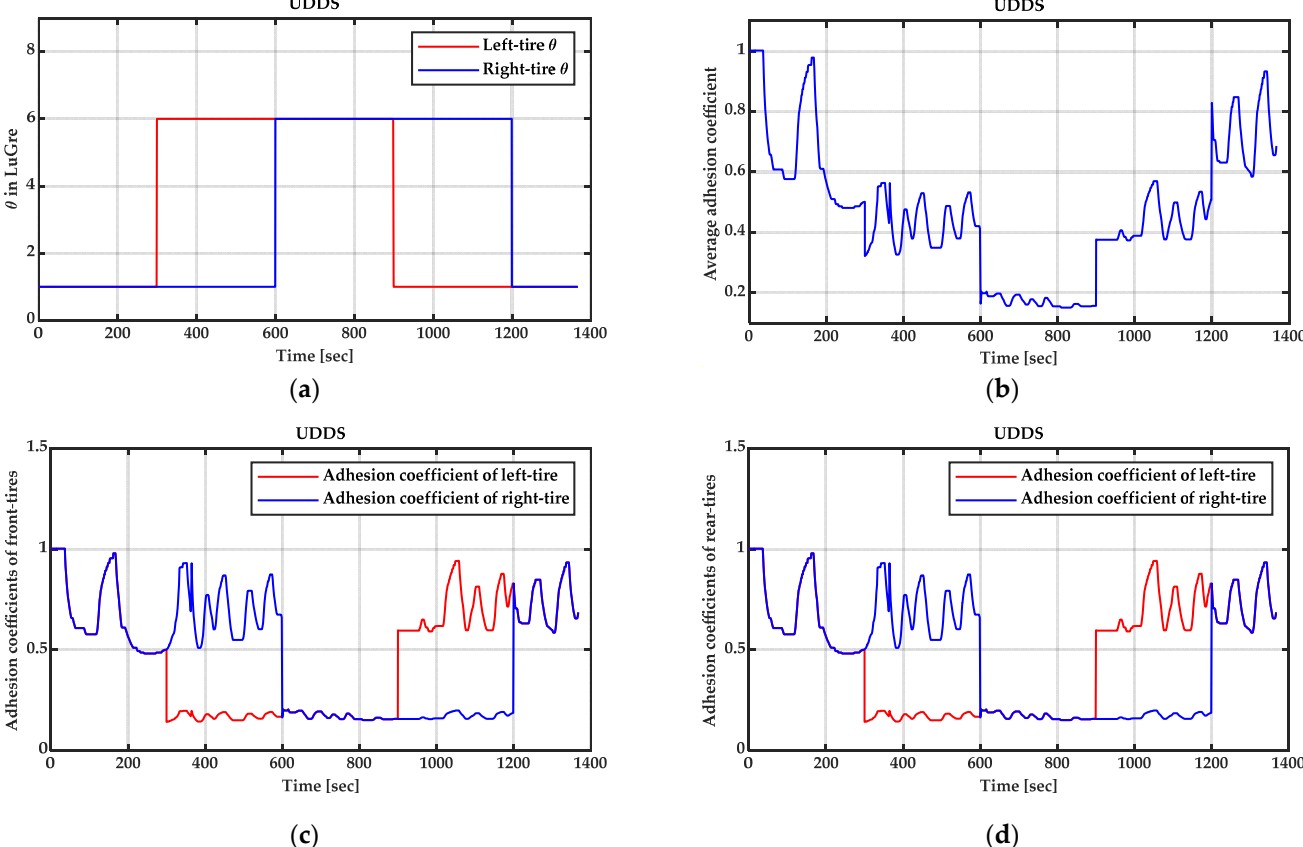

**Figure 7.** *Cont.*

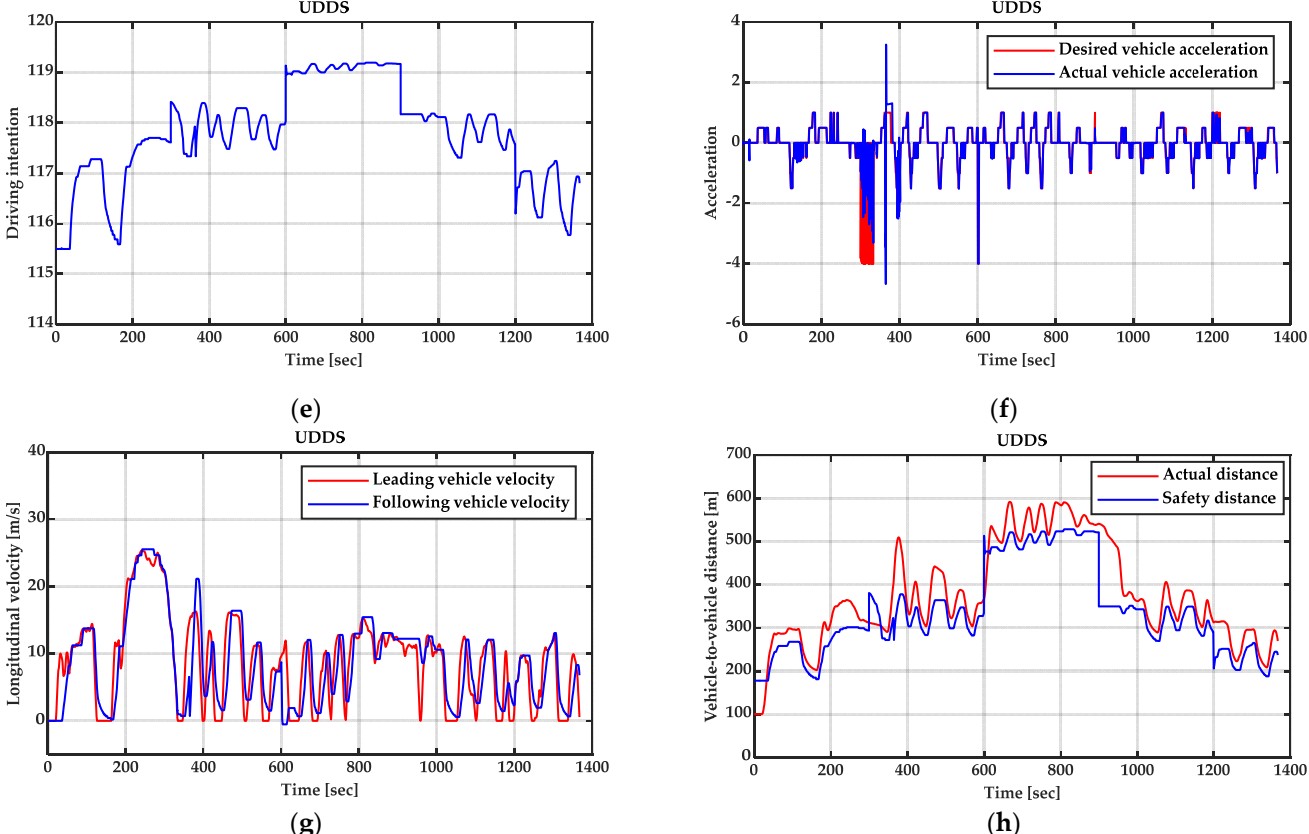

**Figure 7.** Dry asphal$_\mathrm{t}$–snow–ice-mixed road experiments in UDDS with deep neural network. (**a**) Road characteristic parameter of left and right tires. (**b**) Average adhesion coefficient. (**c**) Adhesion coefficients of front tires. (**d**) Adhesion coefficients of rear tires. (**e**) Driving intention. (**f**) Acceleration of following vehicle. (**g**) Longitudinal velocities of leading and following vehicles. (**h**) Vehicle-to-vehicle distance and actual distance.

3.2.2. HWFET Simulation Experiments

HWFET was also selected as the simulation highway condition in this work. The cycle time was 765 s; distance traveled was 16.51 km; maximum speed was 96.4 km/h; average speed was 77.58 km/h; maximum acceleration was 1.43 m/s$^2$; maximum deceleration was −1.48 m/s$^2$; and number of stops was 1 [2,5,28].

(1) Snow and ice road simulation experiments

Road characteristic parameter $\theta$ can be also set to the same values, namely $\theta = 6$, for the left and right tires to represent a snow and ice road condition. As seen in Figure 8a, the road conditions of the left and right tires were the same. The average adhesion coefficient, which is shown in Figure 8b, was calculated by four adhesion coefficients from the LuGre dynamic friction models of the four tires. With the average adhesion coefficient, driving intention $k_d$ can be inferred by the deep neural network. The calculated adhesion coefficient can vary between 0.0 and 0.2 due to the change of the PMSM speed. The average adhesion coefficient provides road surface information for the deep neural network to infer driving intention. Compared to the change trends in Figure 8b,c, the variation trend of driving intention is opposite to that of the average adhesion coefficient. That is to say, when the average adhesion coefficient is small, the driving intention is large, and vice versa; it also conforms to drivers' driving habits. Acceleration closed-loop curves are shown in Figure 8d and actual acceleration can basically track the desired acceleration in the whole driving process. Longitudinal velocities of the leading and following vehicles are shown in Figure 8e, and the vehicle-to-vehicle distance and safety distance are shown in Figure 8f. The actual vehicle-to-vehicle distance is greater than the safe distance and the safe distance is greater than 480 m. This demonstrates that electric vehicles can drive safely and the

proposed driving intention safety distance model is reasonable, feasible, and effective in HWFET.

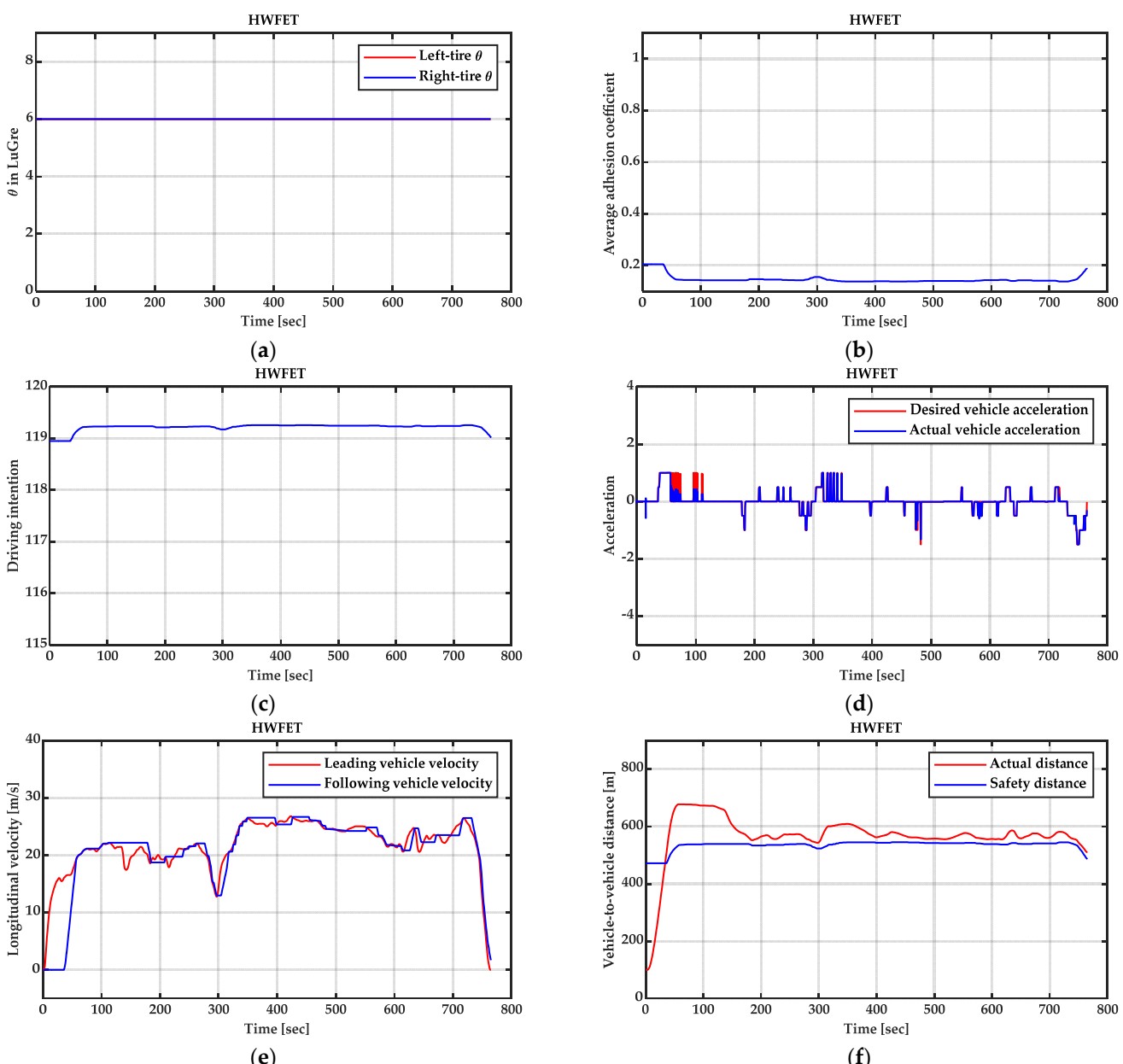

**Figure 8.** Snow and ice road experiments in HWFET with deep neural network. (**a**) Road characteristic parameter of left and right tires. (**b**) Average adhesion coefficient. (**c**) Driving intention. (**d**) Acceleration of following vehicle. (**e**) Longitudinal velocities of leading and following vehicles. (**f**) Vehicle-to-vehicle distance and actual distance.

(2) Dry asphalt–snow–ice-mixed road simulation experiments

Road characteristic parameter $\theta$ can be also set to different values in order for the left and right tires to represent the different road conditions. As seen in Figure 9a, the road conditions of the left and right tires had the same situation but also different situations. The road conditions of the left and right tires included a dry asphalt road as well as a snow and ice road. Therefore, they can represent all the road conditions encountered by tires. The average adhesion coefficient, which is shown in Figure 9b, was calculated by four adhesion coefficients from the LuGre dynamic friction models of the four tires. With the average adhesion coefficient, driving intention $k_d$ can be inferred by the deep neural

network. The calculated adhesion coefficient can vary between 0.0 and 1.0 due to the change of the PMSM speed. Figure 9c,d show the adhesion coefficients of the four tires. The adhesion coefficients of the left tires are the same as that of the right tires. Furthermore, the values of the adhesion coefficients of the four tires cover all road conditions, namely a dry asphalt road as well as a snow and ice road. They can provide adequate road surface information for the deep neural network to infer driving intention. Compared to the change trends in Figure 9b,e, the variation trend of driving intention is opposite to that of the average adhesion coefficient. That is to say, when the average adhesion coefficient is small, the driving intention is large, and vice versa; it also conforms to drivers' driving habits. Acceleration closed-loop curves are shown in Figure 9f and actual acceleration can basically track the desired acceleration in the whole driving process. The actual acceleration curve fluctuated once at about 180 s, which was mainly caused by the road condition changes of the left and right tires. The fluctuation can also immediately disappear due to acceleration closed-loop control. Longitudinal velocities of the leading and following vehicles are shown in Figure 9g, and the vehicle-to-vehicle distance and safety distance are shown in Figure 9h. The actual vehicle-to-vehicle distance is basically greater than the safe distance and the safe distance is greater than 180 m. This demonstrates that electric vehicles can drive safely and the proposed driving intention safety distance model is also reasonable, feasible, and effective in HWFET.

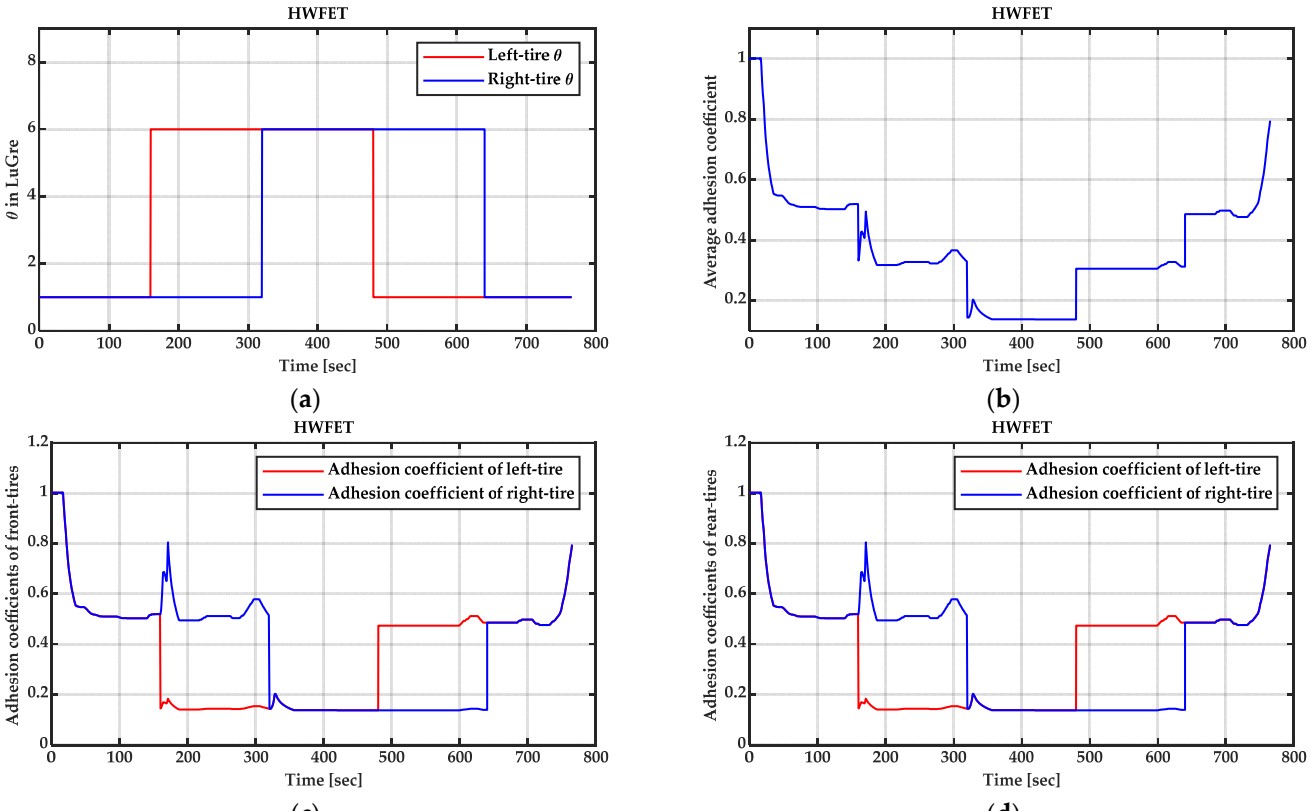

**Figure 9.** *Cont.*

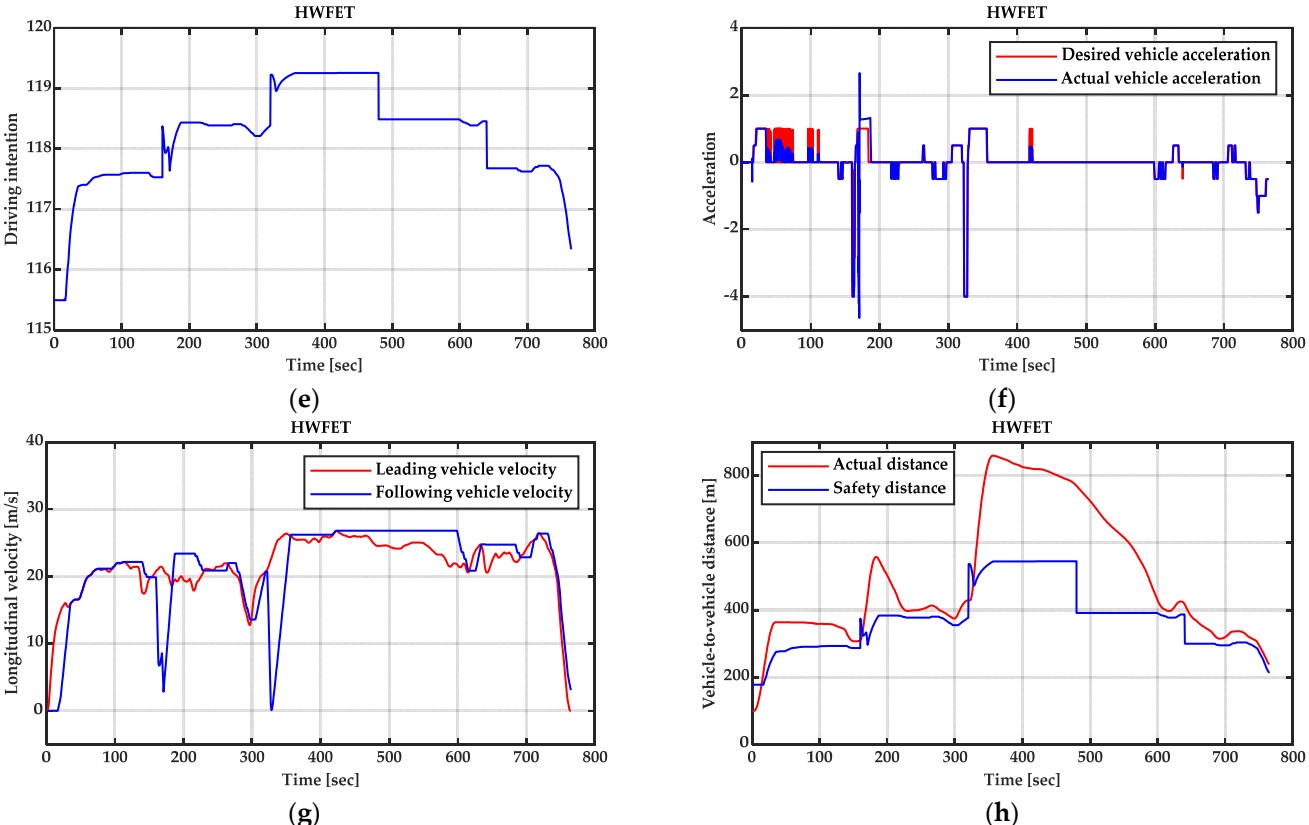

**Figure 9.** Dry asphalt–snow–ice-mixed road experiments in HWFET with deep neural network. (**a**) Road characteristic parameter of left and right tires. (**b**) Average adhesion coefficient. (**c**) Adhesion coefficients of front tires. (**d**) Adhesion coefficients of rear tires. (**e**) Driving intention. (**f**) Acceleration of following vehicle. (**g**) Longitudinal velocities of leading and following vehicles. (**h**) Vehicle-to-vehicle distance and actual distance.

## 4. Conclusions

This paper has presented a driving intention safety distance model based on a deep neural network with dropout regularization in an active collision avoidance system. With the information on the average adhesion coefficient of the four tires, the deep neural network with one input, three hidden, and one output layer can directly infer driving intention. The effectiveness in the control scheme, simplicity in structure, and flexibility in implementation have been verified through simulation experiments under different driving conditions using the RCP and HIL simulator. The proposed driving intention safety distance model can guarantee the safe driving of electric vehicles. In this work, the biases of the nodes were regarded as 0. Therefore, the deep neural network structure is simplified and the computational burden is reduced. Furthermore, driving intention can be calculated by a single control unit, namely the active collision avoidance controller, rather than by an upper-layer network platform.

In addition, to simplify the deep neural network structure, the biases of the nodes were regarded as 0 and the weights were not considered in this work. Although the deep neural network can infer driving intention and the vehicle can drive safely, there is still some deviation between the inferred driving intention and actual driving intention resulting from the ignoring of the bias in the deep neural network. This needs to be further improved. This work will be further improved in future work.

**Author Contributions:** Conceptualization, Y.L. and S.L.; methodology, Z.S.; software, B.L. and J.H.; validation, Y.L., S.L. and Z.S.; formal analysis, Y.L.; investigation, Z.N.; resources, Z.S.; data curation, Y.L. and J.H.; writing—original draft preparation, Y.L.; writing—review and editing, Y.L.; visualization, Y.L.; supervision, Z.N.; project administration, S.L. and B.L.; funding acquisition, S.L. and B.L. All authors have read and agreed to the published version of the manuscript.

**Funding:** This research was funded by the National Natural Science Foundation of China, grant number 62106023, Science and Technology Development Project of Jilin Province, grant number 20210201106GX, and Science and Technology Project of Jilin Province Education Department, grant number JJKH20210746KJ.

**Institutional Review Board Statement:** Not applicable.

**Informed Consent Statement:** Not applicable.

**Data Availability Statement:** Not applicable.

**Conflicts of Interest:** The authors declare no conflict of interest.

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
