# Peer review of "Driving Intention Inference Based on a Deep Neural Network with Dropout Regularization from Adhesion Coefficients in Active Collision Avoidance Control Systems"

_electronics, doi:10.3390/electronics11152284_

Round 1

Reviewer 1 Report

It is an interesting paper that deals with the topic of automatic cruise control under harsh conditions. The paper is well developed, and the result is presented well. The problem is the evaluation of the proposed method lacks quantitative comparison with another method. 

The computational of learning and reaction time is not discussed. Adding a paragraph in this aspect will enhance the quality of this paper.

Author Response

Reviewer: 1#

Comment 1: It is an interesting paper that deals with the topic of automatic cruise control under harsh conditions. The paper is well developed, and the result is presented well. The problem is the evaluation of the proposed method lacks quantitative comparison with another method.

Response 1: In line 128, [2] and [5] are added in the text. In the previous paper published by authors, the driving intention is considered as a constant, such as 1.0 in UDDS, and 10.0 in HWFET. The simulation results show that the constant can also ensure vehicle safety, namely, actual vehicle-to-vehicle distance is basically greater than safe distance. The curve change tends for constant is basically the same as that in this work. Therefore, the simulation results with constant driving intention are not shown in the paper. Besides that, constant driving intention has no ability to adapt to different roads. The driving intention inferred by a deep neural network in this work is changed according to the different road surfaces. Therefore, the simulation results with driving intention calculated by a deep neural network are mainly shown in the paper.

Comment 2: The computational of learning and reaction time is not discussed. Adding a paragraph in this aspect will enhance the quality of this paper.

Response 2: The computational of learning and reaction time are added in the text, and data analysis is also added. This part is marked in blue in line 165-167, and line 172-175.

Special thanks to you for your good comments.

Reviewer 2 Report

The main aim of this manuscript is to proposes a driving intention safety distance model based on a deep neural network. Some conclusions are given. The paper could be interesting but the text cannot be accepted in the present form.  I would suggest a moderate revision after eliminating some shortcomings.

General remark: The figures should represent the main result of the study and their description should give some conclusion of the investigation. It is incomprehensible that the authors put too small Figs. in the text. The conclusion is that then the figures are irrelevant and unnecessary? The authors need to correct that. More precisely they should be enlarged and/or text on their axes or the whole figure.

It might be interesting to put some of the results as a project on github.

Why introduce abbreviation like ADAS, RPC, ... in Abstract when they appear only ones.

The introduction needs to be corrected i.e. rewritten. This Section should be divided into paragraphs. At the moment, some are too long and the reader is tired of following the meaning

The part of the text from line 58-75 should be rewritten. The sentences are almost taken in its entirety from the papers doi:10.1109/TSMCC.2012.2198212, 10.1109/TVT.2020.3011672, 10.1109/ICTIS.2017.8047760

The whole conclusion is general and needs to be reworked, ie. be more specific with statements.

Authors stated “In addition, to simplify the deep neural network structure, the biases of the nodes are regarded as 0, the weights are considered in this work.” Can the author further elaborate this?

Literature is appropriate.

Author Response

Reviewer: 2#

Comment 1: The figures should represent the main result of the study and their description should give some conclusion of the investigation. It is incomprehensible that the authors put too small Figs. in the text. The conclusion is that then the figures are irrelevant and unnecessary? The authors need to correct that. More precisely they should be enlarged and/or text on their axes or the whole figure.

Response 1: The figures in the text, special Figure.6-Figure.9, are zoomed, and the text captains are marked in each figure with two lines.

Comment 2: It might be interesting to put some of the results as a project on github.

Response 2: We will accept expert’s advice to put some of the results as a project on github.

Comment 3: Why introduce abbreviation like ADAS, RPC, ... in Abstract when they appear only ones.

Response 3: ADAS is an advanced driver assistant system, which can assist drivers to complete the necessary driving behavior to prevent traffic accidents from happening in  emergencies. And the active collision avoidance control system mentioned in this work is a kind of ADAS. Therefore, ADAS is introduced in Abstract and Introduction, respectively.

RCP and HIL are experimental equipment in simulation platform. In this work, RCP is MicroAutoBoxâ…¡, this part is added in line 176, namely, “The hardware platform is shown in Figure 4, MicroAutoBoxâ…¡, which is rapid control prototyping (RCP), is used as the simulator of active collision avoidance controller. ”

Comment 4: The introduction needs to be corrected i.e. rewritten. This Section should be divided into paragraphs. At the moment, some are too long and the reader is tired of following the meaning.

Response 4: The introduction is divided into two parts: Motivation and Related Works.

Comment 5: The part of the text from line 58-75 should be rewritten. The sentences are almost taken in its entirety from the papers doi:10.1109/TSMCC.2012.2198212, 10.1109/TVT.2020.3011672, 10.1109/ICTIS.2017.8047760

Response 5: At the request of reviewer, the part of the text from line 58-75 is rewritten except the methods described in the original paper. This part is revised as follows: “An intention predication model based on attribute-Driven Hidden Markov Model Trees is proposed for intention prediction [9]. An efficient recognition approach based on Nonlinear Polynomial Regression (NPR) and Recurrent Hidden Semi-Markov Model (R-HSMM) is proposed to recognize the driver lane-change intention accurately in the early stage [10]. HMMs can only focus on the current driving behaviors rather than past driving behaviors, although they are widely used for driving intention inference. Fuzzy reasoning methods are a class of experience models, therefore, they can get a well quantified driving intention. A fuzzy reasoning-based longitudinal minimum safety distance model is designed with the information on driver's intention and driving circumstance [11]. A fuzzy logic inference system is applied to identify driving intention for hybrid vehicles. The membership functions and rules of fuzzy logic inference system are built for intention identification, and the simulation is done in different driving conditions [12]. ”

Comment 6: The whole conclusion is general and needs to be reworked, ie. be more specific with statements.

Response 6: The whole conclusion is rewritten.

Comment 7: Authors stated “In addition, to simplify the deep neural network structure, the biases of the nodes are regarded as 0, the weights are considered in this work.” Can the author further elaborate this?

Response 7: Due to the authors’ negligence, this sentence is wrong. It can be modified as “In addition, to simplify the deep neural network structure, the biases of the nodes are regarded as 0, the weights are not considered in this work. ”.

Special thanks to you for your good comments.

Reviewer 3 Report

The authors presented a driving intention safe distance model based on a deep neural network with dropout regularization applied in an active collision avoidance system. With information about the average adhesion coefficient of four tires, a deep neural network with a 1-in-3-hidden-1-output layer can directly infer driving intentions. And it was verified through simulation experiments under various operating conditions using RCP and HIL simulators. In particular, the driving intention safe distance model proposed by the authors can ensure safe driving of electric vehicles.

In this study, the use of Deep Neural Network from a vehicle control point of view for ADAS is considered to be suitable for the recent trend. However, as the author mentioned, the limitations of this study should be improved with future research. It is expected to be applied and verified in real vehicles in the future.

As it is written in an easy-to-understand manner, there are no separate points to point out.

It would be nice if you could correct one thing.

1. Regarding the sentence written in the abstract, "Driving intent can be inferred via deep neural networks through dropout regularization from the coefficient of adhesion between tire and road" sounds somewhat abstract. It should be written more logically within the text.

Author Response

Reviewer: 3

Comment 1: Regarding the sentence written in the abstract, "Driving intent can be inferred via deep neural networks through dropout regularization from the coefficient of adhesion between tire and road" sounds somewhat abstract. It should be written more logically within the text.

Response 1: At present, driving intention can be mainly obtained by deep neural network with neuromuscular dynamics and electromygraphy (EMG) signals of drivers. This method needs numerous drivers’ signals and neural network with complex structure. This paper proposes a driving intention direct inference method, namely, direct inference from road surface condition. This part is added in the abstract.

Special thanks to you for your good comments.

Round 2

Reviewer 2 Report

The authors  improved the text. They could have put a little more effort into it but the revised version has met the conditions for acceptance.